# Chemoimmunotherapy in Advanced Biliary Tract Cancers: A Meta-Analysis of Clinical Outcomes

**DOI:** 10.3390/biomedicines13092099

**Published:** 2025-08-28

**Authors:** Alireza Tojjari, Sepideh Razi, Osama M. Younis, Ramez M. Odat, Ibrahim Halil Sahin, Anwaar Saeed

**Affiliations:** 1Department of Medicine, Division of Hematology & Oncology, University of Pittsburgh Medical Center (UPMC), Pittsburgh, PA 15213, USA; alirezatojjari@gmail.com (A.T.);; 2Research Center for Immunodeficiencies, Children’s Medical Center, Tehran University of Medical Sciences, Tehran 1416753955, Iran; 3School of Medicine, The University of Jordan, Amman 11942, Jordan; 4Faculty of Medicine, Jordan University of Science and Technology, Irbid 22110, Jordan; 5UPMC Hillman Cancer Center, 5150 Centre Avenue, Pittsburgh, PA 15213, USA

**Keywords:** biliary tract cancer, chemoimmunotherapy, immune checkpoint inhibitors, meta-analysis, chemotherapy, immunotherapy

## Abstract

**Background/Objectives**: Biliary tract cancers (BTCs), encompassing tumors of the bile ducts, gallbladder, or ampulla of Vater, are notoriously hard to manage, especially when surgery is off the table and standard chemotherapy provides only modest benefits. While emerging treatments such as immune checkpoint inhibitors have shown promise, mixed clinical trial results and varied study endpoints have left their true impact unclear. This concise review consolidates current evidence on combining chemotherapy with immunotherapy to clarify whether these regimens can significantly improve outcomes and steer more effective treatment strategies for BTCs. **Methods:** A comprehensive literature search was conducted across PubMed, Embase, Web of Science, and ClinicalTrials.gov for randomized controlled trials (RCTs) and prospective comparative studies published from January 2010 to December 2024. Fixed-effect meta-analyses (inverse-variance method) were used as the primary approach, with random-effects models (REML) performed as sensitivity analyses to confirm robustness were performed to calculate pooled hazard ratios (HRs) for overall survival (OS) and progression-free survival (PFS). Leave-one-out sensitivity analyses and Egger’s tests assessed result stability and publication bias. The review was conducted in accordance with PRISMA 2020 guidelines and registered in OSF. **Results:** Two RCTs (*n* = 1754; chemoimmunotherapy *n* = 874, chemotherapy *n* = 880) were included in the quantitative meta-analysis. Compared to chemotherapy alone, chemoimmunotherapy significantly reduced the risk of death by 20% (OS, HR = 0.80; 95% CI 0.72–0.89; I^2^ = 0%) and the risk of disease progression or death by 19% (PFS, HR = 0.81; 95% CI 0.73–0.90; I^2^ = 33.5%). Leave-one-out sensitivity analyses confirmed result stability. Egger’s tests showed no significant publication bias (OS *p* = 0.30; PFS *p* = 0.40). Two additional studies (IMbrave 151 and Monge 2022) lacking comparative survival data were qualitatively assessed. **Conclusions**: Chemoimmunotherapy significantly improves OS and PFS compared with chemotherapy alone in advanced BTC, with consistent findings across included trials. These results support the incorporation of chemoimmunotherapy as a first-line therapeutic strategy. Future research should prioritize biomarker-driven patient selection, evaluation of long-term clinical outcomes, and integration of targeted therapies with chemoimmunotherapy.

## 1. Introduction

Biliary tract cancers (BTCs) include malignancies that arise from the intrahepatic and extrahepatic bile ducts, the gallbladder, and the ampulla of Vater. They are usually diagnosed at advanced stages when curative interventions are no longer possible [1]. Due to their aggressive biology and late presentation, BTCs are rarely eligible for surgical resection and have traditionally been managed with systemic chemotherapy. However, gemcitabine–cisplatin regimens, the longstanding standard, offer only modest survival benefits, underscoring the urgent need for more effective approaches [2,3,4]. Key late-phase trials include KEYNOTE-966 [5], TOPAZ-1 [6], and the phase II durvalumab ± tremelimumab study [7] evaluating gemcitabine–cisplatin plus checkpoint inhibitors.

In recent years, immune checkpoint inhibitors have spurred a paradigm shift in oncology, prompting their investigation in BTCs. Preclinical data support synergy between chemotherapy and programmed cell death protein 1 (PD-1)/programmed death-ligand 1 (PD-L1) blockade through enhanced antigen presentation and immune cell infiltration, laying the groundwork for clinical exploration [8,9].

This rapid evolution in treatment modalities marks a significant shift toward personalized medicine in managing BTCs, enabling clinicians to tailor regimens based on tumor biology and patient characteristics. Despite promising mechanistic insights, heterogeneous outcomes across early-phase and pivotal trials underscore the need for a comprehensive evaluation of chemoimmunotherapy strategies. This meta-analysis systematically synthesizes data from randomized controlled trials (RCTs) and prospective comparative studies published between 2010 and 2024 to assess the effect of adding immunotherapy to chemotherapy versus chemotherapy alone on survival outcomes in patients with advanced BTCs. Such insights are critical for informing clinical practice and shaping future research in this challenging disease.

## 2. Materials and Methods

### 2.1. Literature Search

A comprehensive literature search was conducted in PubMed, Web of Science, ClinicalTrials.gov, and the Cochrane Library for studies published between 1 January 2010, and 31 December 2024. Search strategies combined terms for “biliary tract cancer” and “chemotherapy” and “immunotherapy” (see Appendix A for detailed strings). We applied the following filters: English-language articles only, human studies, and clinical trials. No other date, age, or gender restrictions were imposed. This systematic review and meta-analysis were conducted in accordance with the Preferred Reporting Items for Systematic Reviews and Meta-Analyses (PRISMA) 2020 guidelines (Figure 1). The review protocol was registered in the Open Science Framework (OSF) under the registration DOI: https://doi.org/10.17605/OSF.IO/5KYJ7.

### 2.2. Inclusion and Exclusion Criteria

The inclusion criteria for this meta-analysis required RCTs or prospective comparative studies enrolling adult patients with unresectable or metastatic BTCs, including intrahepatic and extrahepatic cholangiocarcinoma, gallbladder cancer, and ampullary carcinoma. Eligible studies had to compare chemotherapy combined with immunotherapy versus chemotherapy alone. To be included, studies were required to report at least one time-to-event survival outcome, overall survival (OS) as the primary outcome, and/or progression-free survival (PFS) as a secondary endpoint.

The exclusion criteria excluded papers from the meta-analysis that studied resectable BTCs, lacked a direct comparison between chemoimmunotherapy and chemotherapy alone, or did not report the primary outcome. Lastly, non-randomized trials, retrospective studies, case reports, editorials, and review articles were all excluded. For qualitative synthesis, studies not meeting the meta-analysis criteria, such as those lacking a chemotherapy-alone control arm or quantitative survival data, but providing relevant qualitative data on chemoimmunotherapy, were included.

### 2.3. Data Extraction and Quality Assessment

Title and abstract screening were independently conducted by two reviewers, with discrepancies resolved by a third reviewer. Full-text screening was also conducted independently by the same two reviewers. Study quality was assessed using the Cochrane Risk of Bias 2.0 (ROB-2) [10] tool and was conducted independently by two separate reviewers. The following data were extracted from each included study: study identifiers (author, year, country), patient and disease characteristics (tumor type, prior treatment, age, sex distribution, sample size), treatment arms (experimental and control regimens), response outcomes (objective response rate [ORR], complete response [CR], partial response [PR], stable disease [SD], progressive disease [PD], disease control rate [DCR]), and time-to-event outcomes (OS and PFS). For the three-arm Phase II trial by Oh et al. [7], hazard ratios (HRs) from the two experimental arms were combined using a fixed-effects variance-based method to avoid double-counting the shared control group. Certainty of evidence (e.g., using the GRADE approach) was not formally assessed due to the limited number of included randomized controlled trials.

### 2.4. Statistical Analysis

All analyses were performed in R (version 4.3.3) using the meta and metafor packages. For OS and PFS, pooled HRs with 95% confidence intervals (CIs) were estimated using a fixed-effect model (inverse-variance method), given the limited number of trials. Between-study heterogeneity was assessed via I^2^ and Cochran’s Q test (with corresponding *p*-values reported for heterogeneity only). The pooled analysis was repeated under a random-effects model using the restricted maximum-likelihood (REML) estimator to confirm the robustness of the fixed-effect results. Leave-one-out influence diagnostics and contour-enhanced funnel plots (with Egger’s regression test) were conducted to assess the impact of individual studies and potential publication bias. Objective response rates, CR, PR, SD, and PD were pooled under a fixed-effect model after Freeman–Tukey double-arcsine transformation, then subjected to leave-one-out sensitivity checks and a random-effects REML model as a robustness check. Statistical significance was defined as *p* < 0.05 (two-sided). Heterogeneity was assessed separately using Cochran’s Q test.

## 3. Results

We identified 61 records through database searches. After removing 2 duplicates, 59 titles and abstracts were screened; 46 were excluded (20 non-original studies, 4 conference abstracts, 23 not meeting PICO criteria). Thirteen full-text articles were assessed for eligibility: eight were excluded (3 non-randomized designs, 4 retrospective cohorts, and one review), leaving four studies for qualitative synthesis. Of these, two RCTs [5,6] provided complete time-to-event data and were included in the quantitative meta-analysis (Table 1).

### 3.1. Baseline Demographic and Clinical Characteristics

Table 1 summarizes the baseline characteristics of 1889 patients enrolled across four clinical cohorts. In the two phase III trials, KEYNOTE-966 [5] randomized 1069 patients roughly equally between gemcitabine–cisplatin plus pembrolizumab (*n* = 533) and gemcitabine–cisplatin alone (*n* = 536); both arms were majority male (53% vs. 51%) with a median age of 64 years (IQR 57–71 vs. 55–70). Most patients (58–60%) had intrahepatic cholangiocarcinoma, with gallbladder (22%) and extrahepatic (18–20%) disease comprising the remainder. Performance status was good (ECOG 0–1 in >95%), chronic hepatitis B was present in about 31% (hepatitis C in 3–4%), and 68% had PD-L1 expression ≥ 1%. TOPAZ-1 [6] enrolled 685 patients into durvalumab plus gemcitabine–cisplatin (*n* = 341) or placebo plus gemcitabine–cisplatin (*n* = 344) arms; both arms were 50–51% male, median age 64 years, and had similar disease sites (56% intrahepatic, 19% extrahepatic, 25% gallbladder). ECOG 0–1 rates exceeded 98%, 21–25% had viral hepatitis, and PD-L1 ≥1% in 58–60%. In Oh et al.’s phase II study [7] (*n* = 124 across three experimental regimens), patients were slightly younger (median 61–66 years), 40–57% were male, predominantly had intrahepatic disease (43–67%), they uniformly had an ECOG of 0, and there were low hepatitis rates (4–10%). Finally, Monge 2022’s single-arm cohort [11] (*n* = 11) mirrored these demographics (64% male, median age 65, intrahepatic site in 100%, ECOG 0 in 82%). Across all studies, patients were well balanced with respect to age, sex, disease distribution, and functional status, ensuring the comparability of efficacy and safety outcomes. Notably, elderly patients aged ≥80 years were rare across all trials, limiting the available evidence for this subgroup.

### 3.2. Characteristics of Studies Included in the Quantitative Meta-Analysis

Table 2 outlines the two pivotal phase III, double-blind randomized trials that form the quantitative backbone of our meta-analysis. In 2023, the global KEYNOTE-966 study (5) enrolled 1069 patients with unresectable or metastatic biliary tract cancer across 175 sites, randomizing 533 patients to pembrolizumab plus gemcitabine–cisplatin and 536 patients to gemcitabine–cisplatin alone; the primary endpoint was overall survival, with a median follow-up of 25.6 months (IQR 21.7–30.4). The following year, the TOPAZ-1 trial (6) spanned 17 countries and similarly compared durvalumab plus gemcitabine–cisplatin (*n* = 341) against gemcitabine–cisplatin alone (*n* = 344) in a comparable patient population; overall survival again served as the primary endpoint, with a median follow-up of 23.4 months (IQR 20.6–25.2). Both studies offer high-quality, randomized data on first-line chemoimmunotherapy versus chemotherapy alone in advanced biliary tract carcinoma.

### 3.3. Qualitative Summary of Two Single-Arm/Protocol Studies

Table 3 provides a concise overview of two non-comparative studies that were excluded from the quantitative synthesis due to their single-arm designs. Monge et al. (2022) [11] enrolled 11 patients in a phase II trial of durvalumab plus tremelimumab, reporting an objective response rate (ORR) of 18% and a six-month overall survival of approximately 54%, but, lacking a control arm, it could not inform direct comparisons. Oh et al.’s 2022 open-label phase II study [7] evaluated three experimental sequences—gemcitabine–cisplatin followed by gemcitabine–cisplatin with durvalumab and tremelimumab, gemcitabine–cisplatin plus durvalumab, and a triplet of gemcitabine–cisplatin, durvalumab, and tremelimumab in 124 patients, yielding ORRs between 50% and 73% and a median overall survival of roughly 18 months. Although these results underscore encouraging activity signals for chemo-immunotherapy combinations, their lack of a randomized control arm precluded their inclusion in the pooled meta-analysis.

### 3.4. Qualitative Synthesis

Two studies lacked a standalone chemotherapy-only comparator and were summarized qualitatively (Table 2):

Oh et al. [7]: Phase II single-arm (*n* = 45); ORR 67%, median OS ~18 mo; PFS not directly reported.

Monge [11]: single-arm (*n* = 11); ORR 18%, 6-mo OS 54%.

### 3.5. Risk of Bias Assessment

The Cochrane ROB-2 tool was applied to the two RCTs included. TOPAZ-1 trial [6] showed a low overall risk of bias; however, the KEYNOTE-966 [5] trial demonstrated some concerns specifically in the outcome measurement domain (Figure 2).

### 3.6. Overall Survival

A fixed-effects meta-analysis demonstrated a significant benefit for chemo-immunotherapy versus gemcitabine–cisplatin alone, with a pooled HR of 0.80 (95% CI 0.72–0.89; z = –4.20, *p* < 0.001) (Figure 3). Under a random-effects model using the REML estimator, the pooled HR remained 0.80 (95% CI 0.72–0.90; z = –3.81, *p* < 0.001) (Figure 3), with τ^2^ = 0 and I^2^ = 0%, confirming the robustness of the fixed-effect estimate. Between-study heterogeneity was negligible (I^2^ = 0.0%, τ^2^ = 0, *p* = 0.44). Egger’s test for publication bias was non-significant (*p* = 0.30) (Figure 3).

### 3.7. Subgroup Analysis (OS)

In trial using pembrolizumab, the pooled HR was 0.81 (95% CI 0.70–0.94; I^2^ = 0%; *p* = 0.78), while in this using durvalumab, it was 0.83 (95% CI 0.72–0.96; I^2^ = 0%; *p* = 0.82), indicating consistent benefit across agents.

### 3.8. Leave-One-Out Analysis (OS)

The leave-one-out sensitivity analysis for OS further validated the robustness of the pooled estimate. Sequential omission of each trial yielded HRs ranging from 0.76 (95% CI 0.64–0.91) to 0.83 (95% CI 0.72–0.95), consistently excluding unity, thus confirming the reliability and stability of the OS benefit.

### 3.9. Progression-Free Survival

The pooled fixed-effect HR for PFS was 0.81 (95% CI 0.73–0.90) (Figure 4), indicating a 19% reduction in the risk of progression or death. Under the random-effects model, heterogeneity remained moderate (I^2^ = 33.5%, τ^2^ = 0.0031, *p* = 0.22) as noted below (Figure 4).

### 3.10. Sensitivity Analysis (PFS)

A random-effects meta-analysis using the REML estimator yielded a pooled log-HR of –0.2131 (SE 0.0682), corresponding to an HR of 0.81 (95% CI 0.71–0.92; z = –3.13, *p* = 0.0018). Between-study heterogeneity was moderate (τ^2^ = 0.0031; I^2^ = 33.5%; Q (1) = 1.50, *p* = 0.22), confirming the robustness of the fixed-effect estimate.

### 3.11. Subgroup Analysis (PFS)

Subgroup evaluation of PFS demonstrated that pembrolizumab (KEYNOTE-966) reduced the hazard of progression or death by 18% (HR 0.82, 95% CI 0.68–1.00; I^2^ = 0%), while durvalumab (TOPAZ-1) achieved a 21% reduction (HR 0.79, 95% CI 0.67–0.93; I^2^ = 0%). These concordant results attest to the efficacy of both agents in prolonging PFS.

### 3.12. Leave-One-Out Analysis (PFS)

The leave-one-out sensitivity analysis for PFS confirmed the robustness of the treatment effect. Omitting individual trials sequentially resulted in HR ranging from 0.75 (95% CI 0.64–0.88) to 0.86 (95% CI 0.74–0.99), with all confidence intervals remaining below 1.0. This consistency supports the stability and reliability of the observed benefit for PFS.

### 3.13. Objective Response Rates

The pooled analysis of objective response rates (ORR), comprising complete response (CR) and partial response (PR), demonstrated that chemoimmunotherapy significantly improved the response rate compared to chemotherapy alone. Based on the fixed-effect meta-analysis, the pooled ORR was calculated as follows:Overall ORR: 34.8% (95% CI, 31.8–37.9%)Pembrolizumab arm (KEYNOTE-966): ORR = 28.7% (95% CI, 25.0–32.4%)Durvalumab arm (TOPAZ-1): ORR = 27.1% (95% CI, 22.4–31.8%)

These results indicate that chemoimmunotherapy enhances tumor shrinkage and improves clinical responses, supporting its incorporation into first-line therapy regimens.

Table 4 highlights the distribution of tumor responses across treatment arms in the two pivotal trials. In KEYNOTE-966 [5], the addition of pembrolizumab to gemcitabine–cisplatin yielded a slightly higher complete response rate (2.1% vs. 1.3%) and virtually identical partial response rates (26.6% vs. 27.2%) compared to chemotherapy alone, translating into overall response rates of 28.7% and 28.5%, respectively. Stable disease rates and progressive disease rates were also comparable between arms (46.2% vs. 47.4% for SD; 19.1% vs. 17.9% for PD), suggesting that pembrolizumab primarily shifts the balance toward deeper responses without markedly altering disease stabilization. In TOPAZ-1, the durvalumab combination demonstrated a more pronounced benefit: the overall response rate improved from 19% (CR + PR: 1% + 18%) with gemcitabine–cisplatin alone to 27.1% (2.1% + 25%) with the addition of durvalumab. This was accompanied by a reduction in progressive disease (14% vs. 15%) and a higher proportion of patients achieving stable disease (59% vs. 64%). Taken together, these findings indicate that while both immune-checkpoint inhibitors enhance response depth when added to standard chemotherapy, the magnitude of benefit appears greater with durvalumab in TOPAZ-1, particularly in boosting partial responses and maintaining disease control.

Pooled (fixed-effect) rates were:CR: 1.54% (95% CI 1.00–2.10)PR: 33.3% (95% CI 30.9–35.7)SD: 41.8% (95% CI 39.5–44.1)PD: 20.8% (95% CI 18.9–22.7)

To evaluate the robustness of our pooled response estimates, we performed a leave-one-out sensitivity analysis in which each trial arm was sequentially omitted and the overall complete response (CR), partial response (PR), stable disease (SD), and progressive disease (PD) rates were recalculated. As summarized in Table 5, excluding any single arm—whether pembrolizumab plus gemcitabine–cisplatin (KEYNOTE-966_E), its control (KEYNOTE-966_C), durvalumab plus gemcitabine–cisplatin (TOPAZ-1_E), or its control (TOPAZ-1_C)—resulted in minimal fluctuations in the pooled estimates (CR 1.49–1.60%; PR 33.0–33.6%; SD 41.5–42.0%; PD 20.4–21.2%). These narrow ranges underscore the stability of our meta-analytic findings and suggest that no individual study disproportionately drives the overall response profile.

### 3.14. Publication Bias and Sensitivity Analyses

Funnel plots were symmetric for both OS and PFS, and Egger’s tests were non-significant (OS *p* = 0.30; PFS *p* = 0.40) (Figure 5). Leave-one-out sensitivity analyses confirmed that excluding any single study did not materially alter the pooled estimates. All supplementary materials used for statistical analyses are provided in the Appendix A for transparency and reproducibility.

## 4. Discussion

In total, two RCTs enrolling 1754 patients were included in this meta-analysis. Chemo-immunotherapy significantly improved OS (HR = 0.83, 95% CI: 0.72–0.96; I^2^ = 0.0%, *p* = 0.44) and PFS (HR = 0.80, 95% CI: 0.67–0.96; I^2^ = 33.5%, *p* = 0.22) compared with chemotherapy alone, indicating a 17% reduction in the risk of death and a 20% reduction in disease progression risk with combination therapy. Egger’s test for publication bias was non-significant for both OS (*p* = 0.30) and PFS (*p* = 0.40).

### 4.1. Clinical Implications

Our meta-analysis demonstrates that adding immunotherapy to first-line chemotherapy yields a clear survival benefit for advanced BTCs over chemotherapy alone. Pooled results from randomized trials show a prolongation of median OS to roughly 12–13 months with chemoimmunotherapy, compared to ~10–11 months with chemotherapy alone [5,6]. This translates to a 17% reduction in the risk of death with combination therapy–a clinically meaningful improvement in a historically hard-to-treat disease. Notably, the addition of an immune checkpoint inhibitor doubled the 2-year survival rate in one study (≈24% vs. 11–12% with chemo alone), indicating a subset of patients achieving durable long-term remission. These gains were achieved without a trade-off in safety: immunotherapy did not markedly increase high-grade toxicities, and no new safety signals emerged in the combination arm. In practical terms, this means oncologists can improve survival outcomes by incorporating immunotherapy into first-line regimens without significantly worsening the expected chemotherapy-related side effects. Taken together, our findings confirm the paradigm shift in BTC management, establishing chemoimmunotherapy as a new standard of care for untreated advanced disease [6,13]. Therefore, while this meta-analysis supports the role of chemoimmunotherapy as the new first-line standard, it mainly corroborates the findings of the pivotal RCTs rather than providing novel evidence. More granular analyses with IPD would be required to evaluate the magnitude of benefit in each clinically relevant subgroup.

Beyond the statistical significance, these results carry important implications for daily practice. Until recently, gemcitabine–cisplatin was the only standard, yielding a median OS of under 1 year. The introduction of immunotherapy has broken this plateau, offering new hope in a cancer with an urgent need for better therapies. Physicians should now consider a platinum/gemcitabine plus PD-1/L1 inhibitor combination as first-line therapy for eligible patients. Current evidence indicates that this benefit extends across broad patient subgroups. For example, improved survival was seen regardless of PD-L1 expression level or patient ethnicity, suggesting we should offer chemoimmunotherapy to all patients without restricting it to PD-L1–positive tumors. The combination appears effective in intrahepatic cholangiocarcinoma, extrahepatic cholangiocarcinoma, and gallbladder cancers alike, although one trial noted a somewhat less pronounced benefit in extrahepatic disease. This observation might relate to clinical factors (extrahepatic BTC patients often have more comorbidities and biliary complications), but it underscores that anatomical subtypes may respond differently, a nuance clinicians should be aware of. Overall, the broad efficacy of chemoimmunotherapy means that, for the first time in over a decade, patients with advanced BTC have a new frontline option that significantly improves survival and the chance of long-term tumor control [14].

Mechanistically, the success of chemotherapy–immunotherapy combinations in BTC is biologically plausible and likely synergistic. Biliary tract tumors are typically immunologically “cold,” failing to elicit strong native immune responses [14]. Chemotherapy can favorably alter this tumor microenvironment: Gemcitabine and cisplatin induce immunogenic cell death, release tumor antigens, and deplete immunosuppressive cell populations. For instance, gemcitabine has been shown to enhance dendritic cell function and cross-prime CD8^+^ T-cells against tumor antigens [15]. Such effects can convert a cold tumor into a more immune-reactive one, essentially priming the patient’s immune system. Adding a PD-1/L1 blocker then amplifies this anti-tumor immunity by unleashing T-cell responses that chemotherapy alone could not sustain. Clinically, this synergy is reflected in our findings: while objective response rates with chemoimmunotherapy were similar to chemotherapy alone in trials, responses lasted longer with the addition of immunotherapy (e.g., median response duration ~9.7 vs. 6.9 months) [15]. This prolonged disease control in responders likely underpins the observed OS benefit despite only modest improvements in median PFS. In summary, the chemoimmunotherapy approach not only attacks the tumor with cytotoxins but also engages the patient’s immune system for a more durable defense–a dual mechanism of action that is now clinically validated in BTC.

### 4.2. Clinical Considerations

Although the included trials were broadly representative of advanced BTC populations, patients aged ≥80 years were rarely enrolled. As a result, the applicability of chemoimmunotherapy in very elderly or frail patients remains uncertain. While gemcitabine–cisplatin has been used with dose adjustments in older adults, the tolerability of adding immune checkpoint inhibitors to chemotherapy in this subgroup requires prospective evaluation. Future studies should incorporate geriatric oncology frameworks and real-world cohorts to clarify efficacy and safety in this underrepresented population.

Another important limitation is the lack of data beyond two years of follow-up. Both KEYNOTE-966 and TOPAZ-1 demonstrated durable survival benefits at interim analyses, but mature data regarding long-term survival, progression-free survival plateaus, and late immune-related toxicities are not yet available. Understanding outcomes beyond 24 months is essential to determine whether chemoimmunotherapy can achieve sustained disease control or functional cure in a subset of patients. Extended follow-up of existing trials and pooled analyses from real-world registries will be crucial to address this gap.

### 4.3. Study Limitations

Despite the encouraging results, we must acknowledge several limitations of this meta-analysis and the underlying studies. First, the evidence is driven largely by two pivotal trials, TOPAZ-1 and KEYNOTE-966, which had similar designs (gemcitabine/cisplatin with either durvalumab or pembrolizumab vs. chemo alone). The limited number of trials (and total sample size of 1754 patients) restricts our ability to perform extensive subgroup analyses and may overemphasize trial-specific biases. For instance, both trials were sponsored studies with rigorous patient selection, enrolling mostly patients with good performance status (ECOG 0–1) and without severe comorbidities. This selection criterion means the trial populations might not fully represent real-world BTC patients, who often are older or have liver dysfunction, etc. As a result, the generalizability of the findings could be somewhat constrained. Moreover, mature quality-of-life (QoL) data remain unavailable, limiting assessment of whether survival gains translate into meaningful patient benefit. Outcomes in routine practice might be different if patients are less fit than those in trials. Early real-world reports suggest the regimen is feasible outside of trials, but efficacy data in broader populations are still maturing. We also note that publication bias is a potential concern, albeit a minor one here–positive trials are available and were included, whereas any negative or small studies (if they exist) might be unpublished and thus not captured in our analysis. With only two randomized studies, formal tests such as Egger’s test were non-significant, though underpowered due to the limited number of studies. While the qualitative studies [7,11] provided valuable preliminary efficacy data, their lack of comparative controls limited their inclusion in the quantitative meta-analysis. Nonetheless, their encouraging ORR highlights potential activity warranting further randomized comparisons.

Another limitation is the clinical heterogeneity inherent to BTC itself and across trials. BTCs comprise a diverse family of tumors (intrahepatic vs. extrahepatic cholangiocarcinoma vs. gallbladder cancer) with distinct biology and epidemiology. Our meta-analysis combined data across these subtypes; although this boosts statistical power, it introduces heterogeneity. Differences in regional patient demographics (about half of the patients in the trials were from Asia, half from Western countries) could also influence outcomes [13]. For example, etiologies like hepatitis, liver fluke infection, or gallstone disease vary by region and might affect tumor behavior and immune interactions. The trials stratified by primary tumor site and geography, and subgroup analyses so far, show consistently positive effects of immunotherapy across most groups. However, the trend toward a smaller benefit in extrahepatic cholangiocarcinoma hints at unrecognized heterogeneity–perhaps those tumors have a different tumor microenvironment or competing risks. Our analysis could not deeply explore such subgroup differences given the data available. Similarly, variations in trial protocol–e.g., KEYNOTE-966 allowed continuation of gemcitabine beyond 6 months in some cases, whereas TOPAZ-1 followed a more fixed chemotherapy course–might lead to minor outcome differences [14]. We did not detect significant statistical heterogeneity between studies for the primary endpoints, which is reassuring, but the clinical diversity of BTC warrants caution in over-generalizing findings to every patient scenario.

From a methodological standpoint, the lack of individual patient data (IPD) in our meta-analysis is a constraint. We relied on published aggregate outcomes (HRs, survival rates, etc.), which prevents nuanced analyses such as determining how specific patient factors (e.g., age, comorbidities, baseline immune markers) correlate with benefit. Only two randomized controlled trials (RCTs) with similar design and comparable study populations were included, limiting the depth of subgroup analyses. Without access to individual patient data (IPD), it is impossible to assess treatment benefit in clinically relevant subgroups, such as those defined by tumor site, ethnicity, or biomarker status. An IPD meta-analysis in the future could better identify which patients benefit the most or least. Additionally, some important outcomes remain insufficiently reported. Quality-of-life (QoL) data and patient-reported outcomes were not yet fully available from these trials at the time of our analysis. Since adding immunotherapy can be costly and is intended to prolong life, it is crucial to know whether it maintains or improves QoL compared to chemo alone. The absence of mature QoL data is a current gap, though preliminary reports suggest patients tolerate the regimen well; formal QoL analyses will bolster the value proposition of chemoimmunotherapy. Finally, the follow-up duration in published studies, while the longest to date in BTC, is still relatively short for capturing tail-end survival [6]. The OS curves suggest some patients achieve long-term survival beyond 2 years; longer follow-up (3–5 years) is needed to determine if a plateau or cure fraction emerges. Ongoing updates (e.g., a 3-year OS showing continued separation of curves) are awaited to confirm the durability of the benefit.

Another important limitation relates to the lack of data in very elderly patients, particularly those aged 80 years or older. Both pivotal trials predominantly enrolled younger, fitter individuals (ECOG 0–1), limiting generalizability to frailer populations more commonly seen in real-world practice. Consequently, the applicability and tolerability of chemoimmunotherapy in the very elderly remain uncertain and warrant dedicated prospective evaluation. In addition, long-term outcomes beyond two years are not yet well established. Although one included trial demonstrated a doubling of the 2-year survival rate with chemoimmunotherapy versus chemotherapy alone, the durability of this benefit remains unclear given the relatively short follow-up. Longer observation (≥3–5 years) will be critical to determine whether early gains translate into sustained survival benefit or even a plateau suggestive of cure.

Another limitation of this review is that the certainty of evidence across outcomes was not formally graded, which restricts the strength of recommendations derived from these findings.

In summary, while the findings are robust at a population level, uncertainties remain regarding applicability in very elderly or frail patients, durability of long-term benefit, and real-world generalizability.

## 5. Future Directions

Our findings highlight several avenues for future research to optimize patient selection and refine treatment strategies in advanced BTC. A top priority is the identification of predictive biomarkers that can distinguish which patients are most likely to benefit from immunotherapy. Currently, no reliable clinical or molecular markers exist to guide this decision [16]. PD-L1 expression did not predict differential benefit: patients experienced similar OS and PFS improvements with chemoimmunotherapy regardless of PD-L1 status. We therefore need to look beyond PD-L1. Comprehensive tumor profiling (genomic, transcriptomic, and immune microenvironment analysis) could uncover features associated with response or resistance. For instance, exploratory studies suggest certain molecular subtypes (e.g., tumors with TP53/KRAS co-mutations) have higher immune infiltration and might respond better to immunotherapy [15]. Similarly, microsatellite instability-high (MSI-H) or mismatch-repair-deficient BTCs, though rare (~1–2% of BTCs), have shown notable sensitivity to immunotherapy alone, with response rates around 40% in that subgroup [17]. Identifying such subsets prospectively and tailoring therapy accordingly will be a crucial step. In the future, we may refine treatment by intensifying or adding immunotherapy for those predicted to respond, while sparing expense and potential immune toxicities in patients unlikely to benefit (who might be better served by alternative strategies). Translational studies embedded in ongoing trials should collect tumor tissue and blood to search for predictive biomarkers (e.g., T-cell infiltrates, tumor mutational burden, specific cytokine or gene expression signatures). As our meta-analysis shows a benefit in an unselected population, biomarker-driven enrichment could further improve the risk–benefit ratio of chemoimmunotherapy.

Mature quality-of-life (QoL) data from these trials remain essential to ensure that survival benefits translate into meaningful improvements in patient well-being, underscoring the importance of comprehensive patient-centered outcomes in future BTC studies.

Future trials should evaluate chemoimmunotherapy combinations with targeted agents (e.g., FGFR2 or IDH1 inhibitors) and integration with loco-regional treatments (e.g., radioembolization, hepatic arterial infusion) to assess potential synergistic effects.

We also see opportunities to enhance therapeutic strategies, building on the success of chemoimmunotherapy. One approach is to test additional or alternative immunomodulatory agents in combination. Ongoing trials are evaluating whether adding a second checkpoint inhibitor (such as an anti-CTLA-4 antibody) to PD-1 blockade plus chemotherapy can deepen responses, or if novel checkpoints (LAG-3, TIGIT, etc.) might further augment anti-tumor immunity in BTC. The rationale is that dual checkpoint inhibition could unleash complementary arms of the immune system, potentially converting a proportion of current non-responders into responders, though at the cost of higher toxicity, which will need careful assessment. BTCs often harbor actionable mutations (e.g., FGFR2 fusions, IDH1 mutations, HER2 amplifications) that are treated with targeted inhibitors in second-line settings. An open question is how to sequence or combine these with immunotherapy. The potential synergy between targeted therapies and immunotherapy remains an area of active investigation, and prospective combination trials are underway. For example, should a patient with an FGFR2 fusion receive immunochemotherapy first-line or go directly to an FGFR inhibitor? At present, immunotherapy is given to all-comers in the first line, and targeted agents are used on progression [16]. Future trials could explore personalized strategies, such as upfront targeted therapy in selected molecular subgroups with immunotherapy reserved for later, or conversely, adding targeted drugs to the chemoimmunotherapy backbone to see if synergistic efficacy results. Careful research is needed to determine the optimal sequencing of these therapies so that patients can derive maximum benefit over the course of their disease.

Future studies should specifically investigate the efficacy of combining chemoimmunotherapy with targeted agents (e.g., FGFR inhibitors or IDH1 inhibitors) and explore the integration of systemic chemoimmunotherapy with loco-regional treatments (e.g., radioembolization, hepatic arterial infusion chemotherapy) to enhance therapeutic responses and outcomes.

Furthermore, as chemoimmunotherapy becomes established in advanced disease, we should examine its role in earlier disease settings. Adjuvant or neoadjuvant trials are warranted in high-risk resectable BTC to see if adding immunotherapy around surgery can eradicate micrometastatic disease and prevent recurrence. Given the positive results in metastatic disease, it is rational to test similar combinations in the postoperative setting or as induction therapy before surgery for borderline resectable tumors. This could potentially increase cure rates for BTC if successful. Additionally, investigating maintenance strategies is important: in current practice, patients typically stop cisplatin after 6–8 cycles due to cumulative toxicity, but continue immunotherapy for a maximum of two years as per current trial protocols. Is there a role for maintenance single-agent immunotherapy beyond two years in those with ongoing response, or can we stop treatment earlier in certain cases? Prospective studies and long-term follow-up will inform how long to treat with immunotherapy to balance efficacy and toxicity/cost. Another area of interest is combining loco-regional therapies with immunotherapy. Techniques like radioembolization, ablation, or hepatic arterial infusion chemotherapy might provoke additional tumor antigen release or alter immune contexture; their combination with systemic immunotherapy could be synergistic and is an intriguing frontier for clinical trials.

Finally, continued monitoring of patient outcomes and quality of life in the real-world adoption of chemoimmunotherapy will be essential. As more patients receive this combination outside of clinical trials, large registry studies or phase IV trials should collect data on efficacy, toxicity, and QoL to ensure the trial results translate to broader populations. Health economics analyses will also be important, given the high cost of immunotherapies; demonstrating a survival benefit is one aspect, but we must also show that this approach is cost-effective and accessible, especially in regions where BTC incidence is high but resources may be limited. In summary, the advent of chemoimmunotherapy has opened multiple research directions: refining patient selection through biomarkers, exploring new combinatorial regimens, moving immunotherapy into earlier stages, and addressing pragmatic issues of treatment duration and cost. Ultimately, prospective biomarker-driven and combination studies, along with real-world registry data on QoL and cost-effectiveness, will be critical to fully realize the potential of chemoimmunotherapy across BTC subtypes.

## 6. Conclusions

In conclusion, this meta-analysis of 1754 patients demonstrated that adding PD-1/L1 inhibitors to gemcitabine–cisplatin significantly reduced the risk of death by 17% (OS HR = 0.83; 95% CI 0.72–0.96) and the risk of progression or death by 20% (PFS HR = 0.80; 95% CI 0.67–0.96). These results support chemoimmunotherapy as a first-line option for advanced BTC. While further long-term and real-world data are needed to optimize patient selection and management, our findings align with pivotal trials that led to PD-1/L1 approval and mark a major advance in systemic BTC treatment [13].

## Figures and Tables

**Figure 1 biomedicines-13-02099-f001:**
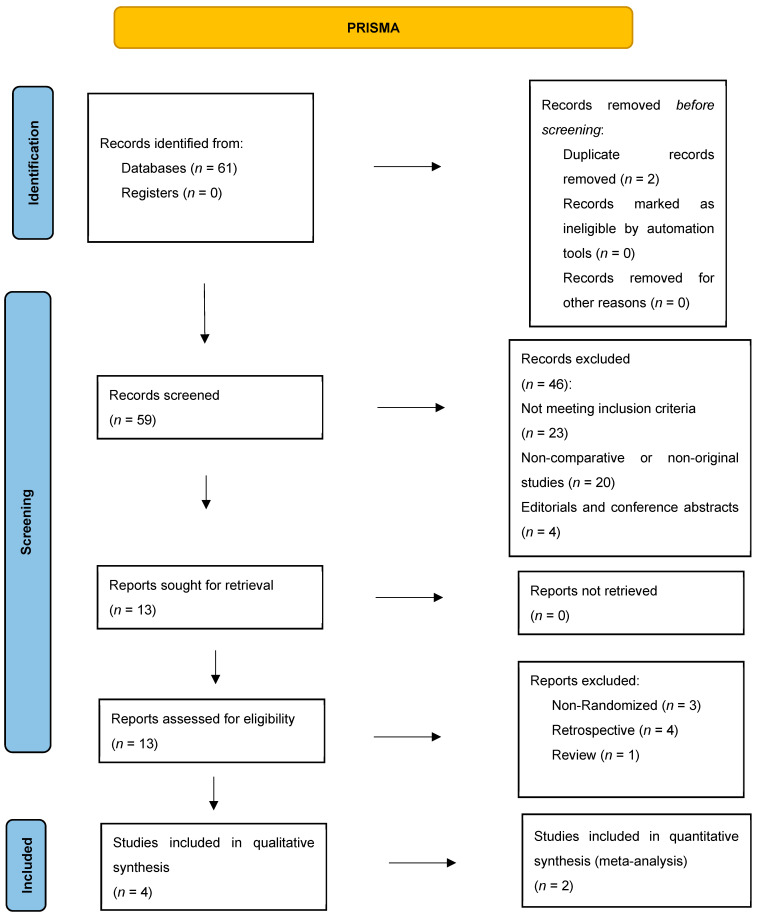
PRISMA flow diagram illustrating the systematic literature search and study selection process. A total of 61 records were identified through database searching; after removing 2 duplicates, 59 records were screened by title and abstract. Of these, 46 were excluded (non-original articles, abstracts, or not meeting PICO criteria), and 13 full-text articles were assessed for eligibility. Eight were excluded (non-randomized or retrospective), leaving four studies: two RCTs included in the quantitative meta-analysis and two studies (single-arm/protocol) included in the qualitative synthesis. The final qualitative and quantitative study selection is shown.

**Figure 2 biomedicines-13-02099-f002:**
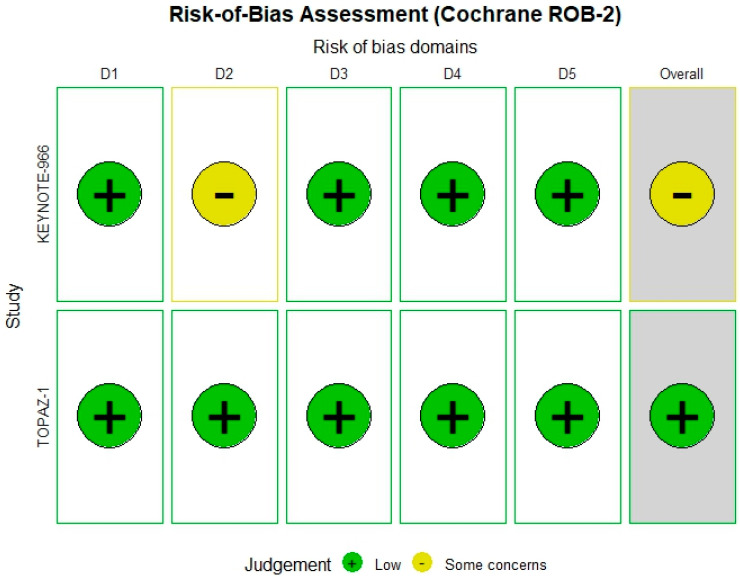
Risk of bias summary according to the Cochrane ROB-2 tool, indicating domain-level assessments for included randomized controlled trials (RCTs).

**Figure 3 biomedicines-13-02099-f003:**
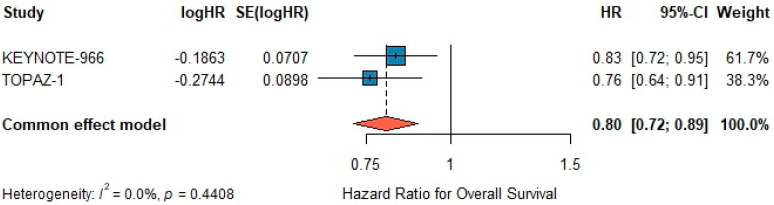
Forest plot for overall survival (OS). Individual hazard ratios (HR) with 95% confidence intervals (CI) for each included study (KEYNOTE-966 and TOPAZ-1) are shown as blue squares. The orange diamond represents the pooled HR calculated using a fixed-effect (common-effect) model (HR = 0.80; 95% CI 0.72–0.89). No significant heterogeneity was detected (I^2^ = 0%, *p* = 0.44). HR < 1 favors chemoimmunotherapy.

**Figure 4 biomedicines-13-02099-f004:**
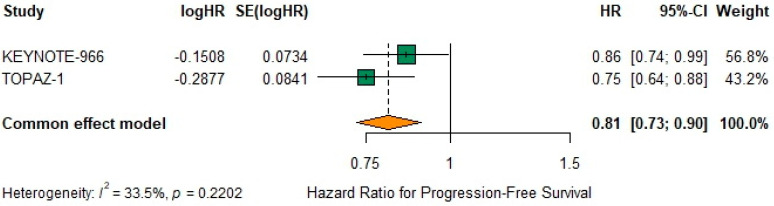
Forest plot for progression-free survival (PFS). Hazard ratios (HR) with 95% confidence intervals (CI) from each study (KEYNOTE-966 and TOPAZ-1) are shown as green squares. The orange diamond represents the pooled HR using a fixed-effect (common-effect) model (HR = 0.81; 95% CI 0.73–0.90). Low-to-moderate heterogeneity was observed (I^2^ = 33.5%, *p* = 0.22), which was not statistically significant. HR < 1 favors chemoimmunotherapy.

**Figure 5 biomedicines-13-02099-f005:**
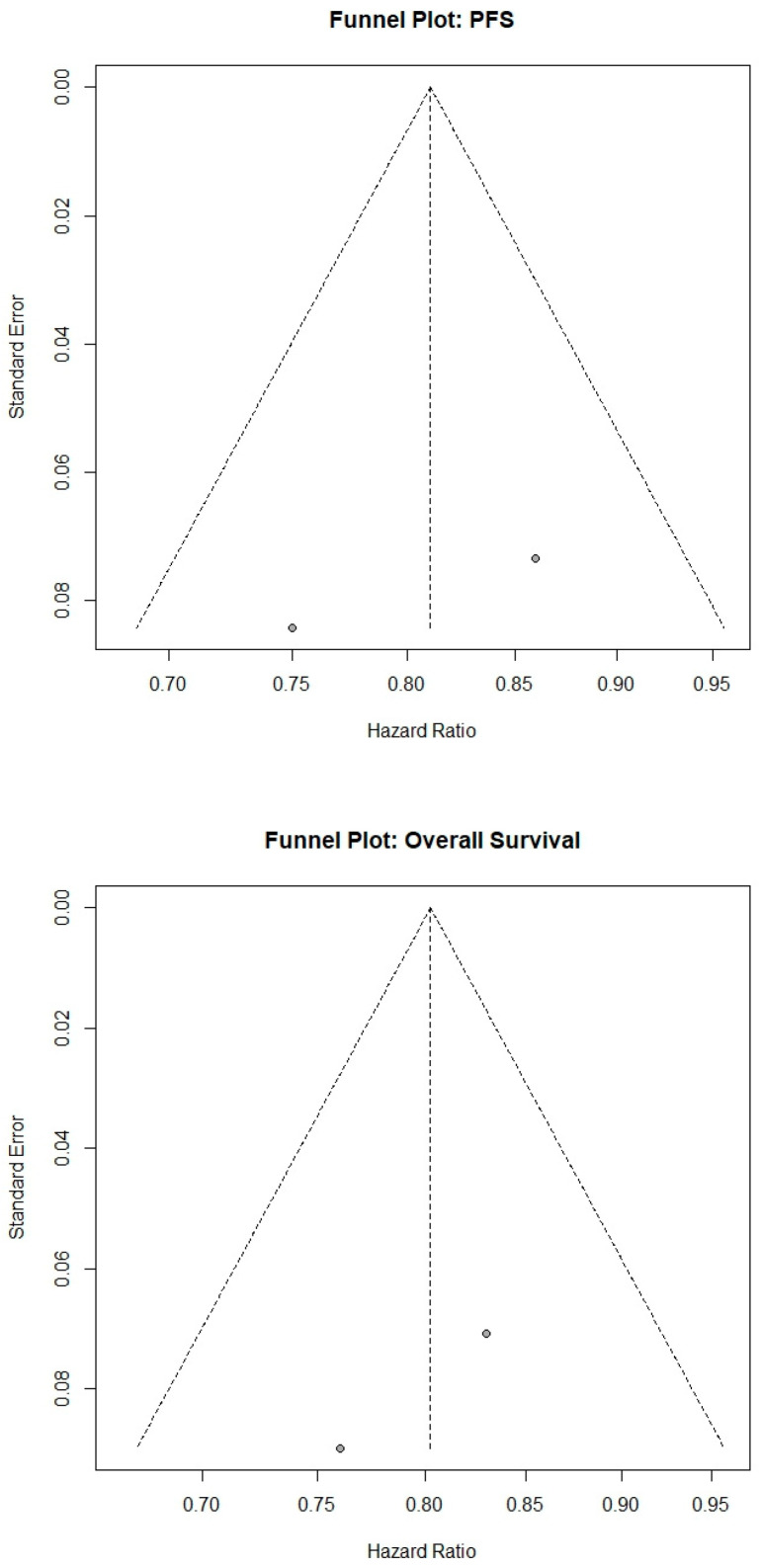
Funnel plots illustrating publication bias assessments for progression-free survival (PFS; upper panel) and overall survival (OS; lower panel). Each circle represents an individual study’s hazard ratio (HR), plotted against its standard error. Visual inspection demonstrates symmetry around the pooled estimates, suggesting minimal publication bias. Egger’s test confirmed no significant publication bias for both PFS (*p* = 0.40) and OS (*p* = 0.30).

**Table 1 biomedicines-13-02099-t001:** Baseline Demographic and Clinical Characteristics of Patients.

Study	Arm	*n*	% Male	Median Age (IQR), yrs	Disease Status (LA/M)	Site	ECOG	Virology (HBV/HCV %)	PD-L1 ≥1 (%)
KEYNOTE-966 ^1^ [5]	GemCis + Pembrolizumab	533	53%	64 (57–71)	60/473	Extrahepatic: 18% Gallbladder: 22% Intrahepatic: 60%	0: 48% 1: 51% ≥2: <1%	HBV: 31% HCV: 4%	68%
	GemCis	536	51%	63 (55–70)	66/470	Extrahepatic: 20% Gallbladder: 22% Intrahepatic: 58%	0: 43% 1: 57% ≥2: 0%	HBV: 31% HCV: 3%	68%
TOPAZ-1 ^2^ [6]	GemCis + Durvalumab	341	50%	64 (20–84)	11/89	Intrahepatic cholangiocarcinoma: 56% Extrahepatic cholangiocarcinoma: 19% Gallbladder cancer: 25%	0: 51% 1: 49%	HBV or HCV: 21%	58%
	GemCis	344	51%	64 (31–85)	17/83	Intrahepatic cholangiocarcinoma: 56% Extrahepatic cholangiocarcinoma: 19% Gallbladder cancer: 25%	0: 47% 1: 53%	HBV or HCV: 25%	60%
Oh et al. (2022) ^3^ [7]	GemCis followed by GemCis + Durvalumab + Tremelimumab	30	57%	64 (57–68)	-	Intrahepatic cholangiocarcinoma: 57% Gallbladder: 23% Extrahepatic cholangiocarcinoma: 7% Ampullary cancer: 13%	0: 100% 1: 0%	HBV: 10% HCV: 0%	NR
	GemCis + Durvalumab	47	40%	61 (57–71)	-	Intrahepatic cholangiocarcinoma: 67% Gallbladder: 15% Extrahepatic cholangiocarcinoma: 19% Ampullary cancer: 4%	0: 40% 1: 60%	HBV: 4% HCV: 0%	NR
	GemCis + Durvalumab + Tremelimumab	47	53%	66 (60–71)	-	Intrahepatic cholangiocarcinoma: 43% Gallbladder: 34% Extrahepatic cholangiocarcinoma: 6% Ampullary cancer: 17%	0: 49% 1: 51%	HBV: 4% HCV: 0%	NR
Monge 2022 ^4^ [11]	GemCis + Durvalumab (single arm)	11	64%	65 (57–72)	100/0	50/50	82	NR	NR

Abbreviations: LA, locally advanced; M, metastatic; IHC, intrahepatic. cholangiocarcinoma; EHC, extrahepatic cholangiocarcinoma; NR, not reported. ^1^ (KEYNOTE-966) [5]: randomized III trial, double-blind. ^2^ (TOPAZ-1) [6]: randomized III trial, open-label. ^3^ Oh et al. [7]: Phase II three-arm study; experimental arms pooled separately. IMbrave 151 [12]: Phase II study protocol only. ^4^ Monge [11]: Phase II single-arm study.

**Table 2 biomedicines-13-02099-t002:** Characteristics of Studies Included in the Quantitative Meta-analysis.

Author (Trial)	Year	Phase/Design	Region	Population	*n* (Exp)	*n* (Ctrl)	Regimen (Experimental vs. Control)	Primary Endpoint	Median Follow-Up (mo)
Kelley et al. (KEYNOTE-966) [5]	2023	III/RCT, double-blind	Global (175 sites)	Unresectable/metastatic BTC	533	536	Gem-Cis + Pembrolizumab vs. Gem-Cis	OS	25.6 (21.7–30.4) *
Oh et al. (TOPAZ-1) [6]	2024	III/RCT, double-blind	17 countries	Unresectable/metastatic BTC	341	344	Gem-Cis + Durvalumab vs. Gem-Cis	OS	23.4 (20.6–25.2)

Data are presented as *n* or median (interquartile range) unless otherwise specified. * Median follow-up is reported as the median (IQR). Abbreviations: RCT, randomized controlled trial; BTC, biliary tract cancer; OS, overall survival; Gem, gemcitabine; Cis, cisplatin.

**Table 3 biomedicines-13-02099-t003:** Qualitative Summary of Two Single-Arm/Protocol Studies.

Author (Trial)	Year	Phase/Design	Sample Size	Intervention Regimen	Comparator	Outcome Summary	Reason for Exclusion
Monge et al. [11]	2022	II/Single-arm	11	Durvalumab + Tremelimumab	None (no control group)	Reported ORR 18%, 6-mo OS ~54%; no comparative outcomes	No control arm; single-arm trial
Oh et al. (Phase II) [7]	2022	II/Single-arm, open-label	124	GemCis followed by GemCis + Durvalumab + Tremelimumab, GemCis + Durvalumab, GemCis + Durvalumab + Tremelimumab	-	ORR 50–73%, median OS ~18 months	Single-arm trial

This table provides a qualitative overview of studies that were excluded from the quantitative meta-analysis due to lack of comparative survival or response data. Oh et al. (2022) are single-arm studies evaluated against historical controls. Abbreviations: RCT, randomized controlled trial; ORR, objective response rate; OS, overall survival; Gem-Cis, gemcitabine plus cisplatin; NR, not reported.

**Table 4 biomedicines-13-02099-t004:** Individual Response Rates (CR, PR, SD, PD) by Study and Treatment Arm.

Study	Arm	*n*	CR *n* (%)	PR *n* (%)	SD *n* (%)	PD *n* (%)
KEYNOTE-966	GemCis + Pembrolizumab	533	11 (2.1)	142 (26.6)	246 (46.2)	102 (19.1)
KEYNOTE-966	GemCis	536	7 (1.3)	146 (27.2)	254 (47.4)	96 (17.9)
TOPAZ-1	GemCis + Durvalumab	341	7 (2.1)	84 (25)	200 (59)	47 (14)
TOPAZ-1	GemCis	343	2 (1)	62 (18)	220 (64)	51 (15)

Abbreviations: CR, complete response; PR, partial response; SD, stable disease; PD, progressive disease; GemCis, gemcitabine plus cisplatin.

**Table 5 biomedicines-13-02099-t005:** Leave-One-Out Sensitivity for Response Rates (CR, PR, SD, PD).

Omitted Study	CR (%) [95% CI]	PR (%) [95% CI]	SD (%) [95% CI]	PD (%) [95% CI]
None (all arms)	1.54 (1.00–2.10)	33.3 (30.9–35.7)	41.8 (39.5–44.1)	20.8 (18.9–22.7)
KEYNOTE-966 _E	1.58 (1.03–2.14)	33.0 (30.6–35.4)	41.9 (39.6–44.2)	20.5 (18.6–22.4)
KEYNOTE-966 _C	1.51 (0.97–2.05)	33.6 (31.1–36.0)	41.7 (39.4–44.0)	21.1 (19.2–23.0)
TOPAZ-1 _E	1.60 (1.05–2.15)	33.2 (30.8–35.6)	41.5 (39.2–43.8)	21.2 (19.3–23.1)
TOPAZ-1 _C	1.49 (0.95–2.03)	33.4 (31.0–35.8)	42.0 (39.7–44.3)	20.4 (18.5–22.3)

Leave-One-Out Sensitivity Analysis for Pooled Response Rates. Each row shows the pooled estimate and 95% confidence interval (CI) for complete response (CR), partial response (PR), stable disease (SD), and progressive disease (PD) after omitting the specified study. Abbreviations: KEYNOTE-966_E, pembrolizumab plus gemcitabine–cisplatin arm of KEYNOTE-966; KEYNOTE-966_C, gemcitabine–cisplatin control arm; TOPAZ-1_E, durvalumab plus gemcitabine–cisplatin arm of TOPAZ-1; TOPAZ-1_C, gemcitabine–cisplatin control arm.

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
