# Peer review of "Chemoimmunotherapy in Advanced Biliary Tract Cancers: A Meta-Analysis of Clinical Outcomes"

_biomedicines, 2025, doi:10.3390/biomedicines13092099_

Round 1
Reviewer 1 Report
Comments and Suggestions for Authors
Thanks to the authors for their work on a really significant topic.
The therapeutic landscape for metastatic biliary tract cancer (BTC) patients has been significantly changing in terms of new treatment options in the last few years. In particular, two phase III randomized trials demonstrated that the combination of standard chemotherapy (CT) with an immune checkpoint inhibitor provides greater efficacy when compared with CT alone, and it is now the new standard of care. Moreover, precision oncology plays a fundamental role in offering innovative targeted therapeutic options to patients progressing to first-line chemoimmunotherapy, as an alternative to standard second-line CT.
The work is methodologically solid, and the ‘Material and Methods’ paragraph is accurate in exposing the inclusion and exclusion criteria of the trials considered. However, only 2 randomized control trials (RCTs) were available for the inclusion in the quantitative meta-analysis, while 2 more single-arm studies were included for the qualitative synthesis.
The presented results are solid and clinically reasonable, with chemo-immunotherapy representing the new standard of care in first-line for BTC patients, with a significant improvement in overall survival and in particular in 2-year survival rate, thanks to a meaningful subset of patients achieving durable long-term remission.
However, considering the specimen of only 2 RCTs, with a similar design and comparable study population, and considering the lack of individual patient data, it is impossible to perform a meaningful subgroup analysis, as stated by the authors.
I’m not sure this meta-analysis actually adds significant data to previously available literature, since it only corroborates the already known indication to chemo-immunotherapy as first-line therapy to all eligible BTC patients.
Having the individual patient data available from the 2 RCTs could allow a more in-depth analysis to really evaluate the depth of benefit in each subgroup.
Reviewer 2 Report
Comments and Suggestions for Authors
This review is of interest, as it integrates the latest evidence regarding the combination of chemotherapy and chemoimmunotherapy for advanced biliary tract cancer and clarifies whether these regimens significantly improve outcomes. Compared with chemotherapy alone, chemoimmunotherapy significantly reduced the risk of death by 20% (OS HR = 0.80, p < 0.001) and the risk of disease progression or death by 19% (PFS HR = 0.81, p = 0.22). Further discussion of both regimens regarding their applicability in elderly patients (especially those aged 80 years or older) and long-term clinical outcomes (≥ 2 years) is warranted.
Round 2
Reviewer 1 Report
Comments and Suggestions for Authors
Thanks to the Authors for taking into consideration my suggestions.
As said before, this work is metholodically solid but doesn't add new evidence to the data already available in the literature. Therefore, specifying that the Authors' objective was to confirm and corroborate the evidence from pivotal RCTs was needed.
As stated, individual patient data would be needed for a more accurate subgroup analysis, so I encourage the Authors to integrate their work with these data, if recoverable.
Also, I appreciated the added observations on the limited follow-up of the trials considered and on the scarce information available on elderly or frailer patients.
Since I believe that no further improvements can be made to the paper, with the inherent limitations adequately acknowledged by the Authors, I'll rely on the judgement of the Editors regarding its final acceptance.
Reviewer 2 Report
Comments and Suggestions for Authors
No additional comments.